# Time Estimation Following an Exhaustive Exercise

**DOI:** 10.3390/jfmk10010035

**Published:** 2025-01-16

**Authors:** Tiziana Maci, Mario Santagati, Grazia Razza, Maria Cristina Petralia, Simona Massimino, Sergio Rinella, Vincenzo Perciavalle

**Affiliations:** 1Department of Mental Health, Alzheimer Psychogeriatric Center, ASP3 Catania, 95127 Catania, Italy; tiziana.maci@aspct.it (T.M.); grazia.razza@aspct.it (G.R.); 2Department of Mental Health, Departmental Module CT2, ASP3 Catania, 95127 Catania, Italy; mario.santagati@aspct.it; 3Department of Clinical and Experimental Medicine, University of Messina, 98122 Messina, Italy; mariacristina.petralia@unime.it; 4Department of Cognitive Sciences, Psychology, Education and Cultural Studies, University of Messina, 98122 Messina, Italy; simona.massimino@unime.it; 5Department of Educational Science, University of Catania, 95125 Catania, Italy; sergio.rinella@hotmail.com; 6Department of Medicine and Surgery, Kore University of Enna, 94100 Enna, Italy

**Keywords:** time, temporal processing, time estimation, exhaustive exercise, blood lactate, healthy adult

## Abstract

**Background/Objectives:** Time estimation was investigated in 24 healthy adults, including 12 women and 12 men, before and after an exhaustive exercise. **Methods**: We compared the ability of estimating time intervals in the range 1 to 5 s using tasks requiring mental counting and tasks that did not allow it. Time estimation and blood lactate levels were evaluated before and at the end of the exercise. **Results**: We found that the perception of time intervals between 1 and 5 s was affected at the end of the exercise. The observed effects, associated with a significant increase in blood lactate levels, were different in the two types of time estimation used in the present study. When participants had to evaluate the duration of the time interval using mental counting, a significant reduction in the overestimation of time made at rest was observed at the end of exercise. On the other hand, when participants had to assess the difference in duration between two events without the possibility of mental counting, a significant deterioration in performance was observed at the end of the exercise. In both cases, no differences were seen between genders. **Conclusions**: It could be hypothesized that an increase in blood lactate, acting as a type of physiological arousal, could contribute to the distortion of perceived time intervals. On the other hand, it does not yet seem possible to propose a model to explain the worsening of the perception of time when mental counting is not possible.

## 1. Introduction

The subjective perception of “time flow” is one of the most important cognitive capabilities in everyday life. Because of its subjectivity, objective time meters, such as sundials, hourglasses, clocks, and calendars, are necessary. French psychologist Paul Fraisse (1911–1996) has suggested that the human processing of time has three functionally distinct components: periods shorter than 0.1 s are perceived as “instantaneous”; periods between 0.1 s and approximately 5 s delimit the “perceived present” or “psychological present”. Periods longer than 5 s are thought to involve long-term memory [1]. Kinsbourne and Hicks [2] defined this period as the “extended present” and estimated its duration to be longer than 30 s.

The data available in the literature seem to indicate that two different modalities are used in the evaluation of time flow. In the first type, which can be defined as “mental” or “verbal” counting, the subject is conscious of the time flow and tries to measure it by mentally counting. It has been observed that, during this type of “mental” timing, which undoubtedly involves working memory [3], there is an involvement of motor structures such as the anterior lobe of the cerebellum of both sides, the supplementary motor area, and the left prefrontal cortex [4]. However, when the subject’s attention is prioritizing another target and mental counting is no longer possible, the subject estimates time flow unconsciously, probably by using the same processes involved in reversal learning, i.e., the adaptation of behavior according to changes in stimulus–reward conditions [5] and/or delay discounting, i.e., how long a person is willing to wait for a reward [6]. Therefore, motor and non-motor aspects seem to be implicated in temporal processing.

There are two main models in the literature describing the process of time perception, the Scalar Expectancy Theory, also called Pacemaker Accumulator Model, and the Striatal Beat Frequency Model [7,8].

The Scalar Expectancy Theory divides the system of measuring time into three successive moments [9]. The process begins with the arrival of a signal that closes a switch and the pulses generated by a Poisson-variable pacemaker are collected in an accumulator. At some point, if an event interrupts the accumulation process, the contents of the accumulator are transferred from working memory to reference memory for long-term storage [7], and the accumulator is reset to zero [10]. When a new event occurs, the stored duration is compared to a new occurrence to determine whether the content of the accumulator at that moment is less than, greater than, or equal to the stored event [7,8,9,10].

The Striatal Beat Frequency Model attempts to identify neural regions involved in the process of timing [11]. The model predicts that in the cerebral cortex, there are cortico-striatal neurons that function as oscillators, i.e., real pacemakers. These neurons send their impulses to striatal spiny neurons that monitor the activity of cortical oscillators. When a target duration is reached, dopamine is released from mesencephalic dopaminergic structures (substantia nigra pars compacta and the ventral tegmental area) onto striatal neurons, which strengthens the cortico-striatal synapses that are active at that moment, and thereby stores the elapsed time interval [12]. Subsequently, when an event occurs, striatal spiny neurons compare the current firing with the stored pattern to identify when the duration has been reached; when the two values match, spiny neurons fire to indicate that the interval has elapsed [11,12,13].

Ever since the pioneering work of Harold Gulliksen [14], an alteration in the perception of time flow after physical activity has been observed [15,16,17,18,19]. However, all these studies have only considered the “mental” mode for time flow assessment. To better understand the role of physical activity in these two different ways of estimating the flow of time, we compared the capability of estimating time intervals from 1 s to 5 s, i.e., in the range of the “present” [1], in subjects performing an exhaustive exercise, by using tasks requiring mental counting, presumably related to motor activation, and tasks that did not allow it. We wanted to verify whether there is a correlation between blood lactate levels and the ability to estimate time flow.

Time perception is an ability strongly associated with various cognitive functions, including attention and memory (see Matthews and Meck [20]).

It has been found that both an exhaustive exercise and a simple intravenous injection of lactate are able to worsen attentional mechanisms [21] and working memory [22].

Since the “non-mental” mode of time estimation occurs in a condition in which the subject’s attention is focused on another activity, it seems possible that this form of time estimation will be most affected by the exhaustive exercise [23]. In fact, it has been observed that this type of exercise is able to inhibit the prefrontal cortex [24] and consequently the cognitive processes that it controls.

## 2. Materials and Methods

*Participants*. The subjects who agreed to participate in the study were 24 healthy adults, aged between 19 and 25 years. Out of these, 12 were women, aged between 19 and 24 years (mean age 22.0 years ± 1.86 SD), with a mean height of 1.66 m (±0.04 SD) and a mean weight of 58.4 kg (±3.73 SD). All the women who participated in the study had a regular menstrual cycle. The remaining 12 volunteers were men aged between 19 and 25 years (mean age 22.4 years ± 1.72 SD), with a mean height of 1.73 cm (±0.03 SD) and a mean weight of 71.8 kg (±5.81 SD). All participants practiced amateur sports for at least one year and had medical authorization to practice non-competitive sport activities. The *t*-test showed no statistically significant differences in age, height and weight between women and men.

The study was approved by the Ethical committee of the Kore University of Enna (number 538, 11 January 2024).

All participants were university students of Sports Science who voluntarily joined the study; they were informed about the trials of the study and the anonymity of their answers before providing their written consent to participate, in accordance with the Declaration of Helsinki.

*Experimental Design*. The tests were carried out between 9 a.m. and 1 p.m. Participants had breakfast before 8 a.m. [25] and had a regular night’s sleep prior to the testing sessions. Particular attention was paid to eliminate, as far as possible, sources of distraction for the subject during the execution of the test. Each subject had to participate in 2 different experimental sessions, as described below:*Session 1*: Each subject performed one of the two time estimation tasks (mental or non-mental, randomly selected) at rest and blood lactate was measured (pre); each subject performed the acute exhaustive exercise, and, subsequently carried out the time estimation task again, with the succession of time sequences randomly modified with respect to the test at rest; blood lactate was measured at the end of the exhaustive exercise, as well as after 5 min and 15 min.*Session 2* (the day after session 1): Each subject performed the second time estimation task (mental or non-mental) at rest and blood lactate was measured (pre); each subject performed the acute exhaustive exercise, and subsequently carried out the time estimation task again, with the succession of time sequences randomly modified with respect to the test at rest; blood lactate was measured at the end of the exhaustive exercise, as well as after 5 min and 15 min.

The overall duration of both tests did not exceed 3 min.

*Exercise*. The participants performed a maximal incremental test on a cycloergomter (Monark, Vansbro, Sweden), at a pedaling rate of 60 rpm, during which the electrocardiogram was monitored. Each subject started with unloaded cycling for 3 min, and the load was increased by 30 W every 3 min until volitional exhaustion or the required pedaling frequency of 60 rpm could not be maintained [22].

*Blood Lactate*. Blood lactate was measured from the fingertip before, immediately after and 15 min after the conclusion of the exercise, using a reliable “Lactate Pro 2” portable lactate analyzer (Arkray Inc, Kyoto, Japan) [26].

*Experiments*. We tested the ability to make temporal discriminations in the range of the psychological “present”, i.e., around 1 to 5 s [1]. To reduce the risk of a learning effect, the subjects did not undergo familiarization trials with the tests before the experimental session. Time estimation was assessed using two different experimental conditions, exploring either quantitative (mental count) or qualitative (comparison of different frames) estimation methods [27]. Figure 1 shows an outline of the two experiments.

*Experiment 1*. In this experiment (Figure 1A), subjects viewed a computer screen where a smiling yellow face appeared. After an interval randomly ranging from 1 to 5 s, a blue square then appeared. The subject was instructed to mentally count for the estimation of the interval between the two visual stimuli and to verbally report to the experimenter the result of their mental calculation. The task was repeated 10 times for each session, so that the subject would evaluate the same time interval twice. The estimation of time was calculated as the percent variation in measured time (MT) with respect to the considered interval (CI), ((MT × 100)/CI) − 100).

*Experiment 2*. In this experiment (Figure 1B), we adapted the task described by Riesen and Schneider [28]. The subjects saw on the computer screen two vertically arranged blue squares (1). Following examiner command, one of the squares turned yellow (2). After a variable period, the second square turned yellow (3). The first one then turned blue again (4), followed by the second (5). Subjects were asked to indicate verbally which square had been yellow for a longer period (the upper or the lower). Within a test series, the interval duration was constantly 1 or 2 s. However, the duration of the yellow phase of the two squares was randomly arranged to obtain differences in duration of 15%, 20%, 25%, 33% or 50%, with 15% indicating the most difficult condition (minimal difference between the two intervals) and 50% being the easiest condition (maximal difference between the two intervals). The number of trials with the first or second interval being longer was equal (the square was 5 times longer for the first interval and 5 times longer for the second interval for each discriminative factor) because it was considered possible that the order might influence the subjects’ responses. Whether the upper or lower square was longer was fully randomized. In this experiment, the number of correct answers was considered as a measurement of ability to estimate time.

*Statistical analysis*. The values obtained were treated statistically using GraphPad Prism version 9.5.1 for Windows (GraphPad Software, San Diego, CA, USA). First, the distribution of data was analyzed using the Shapiro–Wilk Normality Test to confirm normal distribution. Subsequently, data were compared using the paired *t*-test (2-tailed) or 1-way repeated measures analysis of variance (ANOVA) using a non-parametric test (Kruskal–Wallis), followed by Dunn’s Multiple Comparison Test, due to the small sample. Correlations were analyzed as Simple Linear Regressions. Significance was set at *p* < 0.05.

## 3. Results

As can be seen in Figure 2, in both experimental conditions, the blood lactate increased significantly at the end of the exhaustive exercise, remained significantly higher 5 min after the end of the exercise and returned to the pre-exercise values 15 min after its end, without significant differences between women and men.

In the first experiment, blood lactate levels in women increased from 1.6 mmol/mL (±0.25 SD) before the exercise, to 7.2 mmol/L (±1.00 SD) at its end, and in men from (1.6 mmol/L ± 0.26 SD) before the exercise to 7.8 mmol/L (±0.56 SD) at its end. In the second experiment, blood lactate levels in women increased from 1.6 mmol/mL (±0.20 SD) before the exercise, to 7.1 mmol/L (±0.77 SD) at its end, and in men from (1.6 mmol/L ± 0.22 SD) before the exercise to 7.3 mmol/L (±0.85 SD) at its end. It is interesting to note that no statistically significant differences were detected (*t*-test) between the blood lactate values detected in each of the four measurements taken during the first experiment (mental count) and those detected at the corresponding times in the second experiment (non-mental count). It should be noted that in both women and men, blood lactate levels showed no significant differences between those measured at the end of exercise and those measured 5 min after its conclusion.

No statistically significant differences were found between lactate levels measured in women and men.

### 3.1. Experiment 1

During the evaluation of short time intervals, by using mental counting, all participants overestimated the time durations, without significant differences between women and men; for the whole sample, the overestimation was +81.6% ± 25.69 SD before the exercise. However, at the conclusion of the exercise, the overestimation was significantly lower (*p* < 0.001) at +23.3% ± 13.82 SD.

A significant correlation (r^2^ = 0.6379, *p* < 0.0001) was found between the overestimation of short intervals performed by participants and blood lactate levels (Figure 3), without significant differences between genders.

### 3.2. Experiment 2

During the evaluation of short time intervals, without the possibility of using mental counting, it was observed that, for the whole sample, the number of correct answers decreased significantly (*p* < 0.001) after the exhaustive exercise, ranging from 9.1 ± 0.78 SD before the exhaustive exercise to 7.5 ± 1.18 SD at the conclusion. A statistically significant decrease (*p* < 0.05) was also observed in men, with the number of correct answers ranging from 9.3 ± 0.65 SD before the exhaustive exercise to 7.8 ± 1.03 SD at the conclusion, and in women (*p* < 0.05), with the number of correct answers ranging from 8.8 ± 0.83 SD before the exhaustive exercise to 7.1 ± 1.24 SD at the conclusion.

Moreover, even in this second experiment, a significant correlation (r^2^ = 0.3093, *p* < 0.0001) was found between the correct answers performed by participants and blood lactate levels (Figure 4), without gender differences.

In this experiment, participants tended to show a higher probability of incorrectly estimating the time duration the more the difference in the duration of the two visual stimuli decreased, a trend that became statistically significant (*p* < 0.05) at the end of the exhaustive exercise (Figure 5).

## 4. Discussion

In the evaluation of short intervals (between 1 s and 5 s), individuals who had just performed an exhaustive exercise exhibited significant changes in their perception of time. These changes were detectable both when the participants estimated the time interval by counting mentally (mental estimation) and when they evaluated the time interval in a condition where this was not possible (non-mental estimation). During mental counting, the subject is conscious of the time flow and tries to measure it mentally. However, when the subject’s attention is focused on another target and mental counting is no longer possible, the subject estimates the time flow unconsciously, probably by using the same processes involved in the reversal learning, i.e., the adaptation of behavior according to changes in stimulus–reward conditions [5].

It is worth noting that time estimation using mental counting is a procedure of which the subject has awareness and so could be considered an explicit cognitive process, whereas time estimation without mental count is an implicit process. Since there is evidence for dissociation between the neural substrates that support explicit and implicit cognitive processes [29], it can be further suggested that the two “timers” are in different parts of the brain.

In this study, where time intervals between 1 and 5 s were studied, a significant overestimation of time (+81.6% ± 25.69 SD) measured with mental count was detected in subjects at rest, without any significant difference between genders. This observation is in line with what was described by the German physiologist Karl von Vierordt (1818–1884) in what is now called Vierordt’s law; “short” intervals of time tend to be overestimated, and “long” intervals of time tend to be underestimated [30].

The difference in time perception in the two types of experiments used in this research is in line with the observation of a reduction in perceived time as the complexity of a task performed simultaneously increases [31]. In fact, it has been shown that, while attention is devoted only to time estimation, there is an overestimation, whereas there is an underestimation of time when attention is diverted elsewhere [32].

In this study, the marked overestimation of time observed at rest with mental counting was significantly reduced (*p* < 0.0001) at the end of the exhaustive exercise (+23.3% ± 13.83 SD). From these results, it seems that exhaustive exercise slowed down the mental count used to estimate time, greatly reducing the overestimation observed at rest. These observations seem to be in line with those found by Goudini et al. [18] who observed significant underestimates in the evaluation of time intervals of 5, 10, 20 and 30 s after physical fatigue. In addition, the present results are congruent with Graham et al. [33], who showed an underestimation of time (5, 10, 20 and 30 s) after 30 s of isometric exercises with knee extensors at 100%, 60%, and 10% of maximum voluntary contraction with respect to the control condition. The present results are also in line with those observed by Edwards and McCormick [26] who asked participants to estimate the completion of 25%, 50%, 75% and 100% of a 30 s test on Wingate cycles and of 20 min on a rowing ergometer, under various ratings of perceived exertion conditions. They observed that at the 75% and 100% intervals, estimates of time under maximal perceived exertion were significantly shorter than those performed under mild and moderate perceived exertion conditions.

The findings that physical fatigue can modify the subjective perception of time measured with mental counting can be explained in terms of the Scalar Expectancy Theory. In the present study, all participants had a significant increase in blood lactate at the end of an exhaustive exercise. It is interesting to note that the observed levels were well above 4 mmol/L, an arbitrary value called OBLA (Onset of Blood Lactate Accumulation) [34] used in the past to indicate the anaerobic threshold [35].

In the present study, while it can be said that an exhaustive exercise is able to influence the time estimation, it cannot be said that the observed influences are dependent on the increase in blood lactate. However, in previous studies, it has been observed that intravenous infusion of lactate in comfortably seated individuals was able to induce changes equal to those observed in the same people after an exhaustive exercise on both attentional processes [21] and excitability of the motor cortex [36]. Furthermore, it is known that lactate not only acts in the brain as a useful metabolite but also exerts a modulating action on brain cells by activating the G protein-coupled receptor hydroxycarboxylic acid receptor 1 (HCA1), previously referred to as GPR81 or HCAR1 [37].

Therefore, it could be hypothesized that a strong increase in blood lactate, acting as a type of physiological arousal [38,39,40], decreases the speed of the pacemaker, resulting in a decrease in the number of pulses collected in the accumulator [41]. As a result, a rise in blood lactate levels induced by an exhaustive exercise, and the consequent enhanced arousal, could reduce the overestimation of perceived time intervals.

On the other hand, it does not yet seem possible to propose a model to explain the worsening of time estimation when mental counting is not possible.

It should be noted that the present study was conducted on a sample of only 24 subjects and that the participants were 12 women and 12 men aged between 19 and 25 years. Even with these limitations, this study found that the perception of time intervals between 1 and 5 s was affected at the end of an exhaustive exercise.

## 5. Conclusions

This study found that the perception of time intervals between 1 and 5 s was influenced by an exhaustive exercise. The observed effects, associated with a significant increase in blood lactate levels, were different in the two types of time estimation used in the present study. When participants had to evaluate the duration of the time interval by mental counting, a significant reduction in the overestimation of time made at rest was observed at the end of exercise. On the other hand, as we hypothesized, a significant deterioration in performance was observed at the end of an exhaustive exercise when participants had to evaluate the difference in duration between two events without the possibility to mentally count.

In sport, this last observation could be useful in those disciplines that require both the estimation of events’ durations and high attentional processes.

In particular, the estimation of “perceived present”, which for Paul Fraisse has a duration of less than 5 s [1], is critical in decision-making moments [42] of many individual (such as tennis) and team sports (such as football), with phases of high intensity that could alter the ability to estimate time flow.

## Figures and Tables

**Figure 1 jfmk-10-00035-f001:**
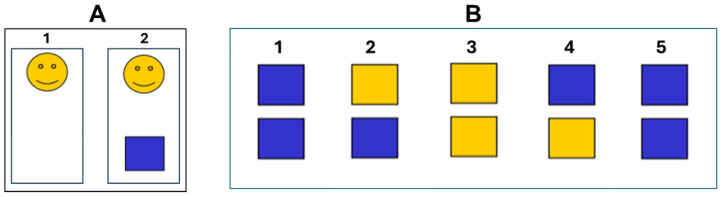
(**A**) summarizes Experiment 1. The subjects viewed a computer screen where a smiling yellow face appeared (1); after an interval randomly ranging from 1 to 5 s, a blue square appeared (2). The subject was instructed to mentally count for the estimation of the interval between the two visual stimuli. (**B**) summarizes Experiment 2. The subjects saw two vertically arranged blue squares on the computer screen (1). Following examiner command, one of the squares turned yellow (2). After a variable period, the second square turned yellow (3). The first one then turned blue again (4), followed by the second square (5). Subjects were asked to indicate verbally which square had been yellow for a longer period (the upper or the lower). Within a test series, the interval duration was constantly 1 or 2 s. However, the duration of the yellow phase of the two squares was randomly arranged to obtain differences in duration of 15%, 20%, 25%, 33% or 50%, with 15% indicating the most difficult condition (minimal difference between the two intervals) and 50% being the easiest condition (maximal difference between the two intervals).

**Figure 2 jfmk-10-00035-f002:**
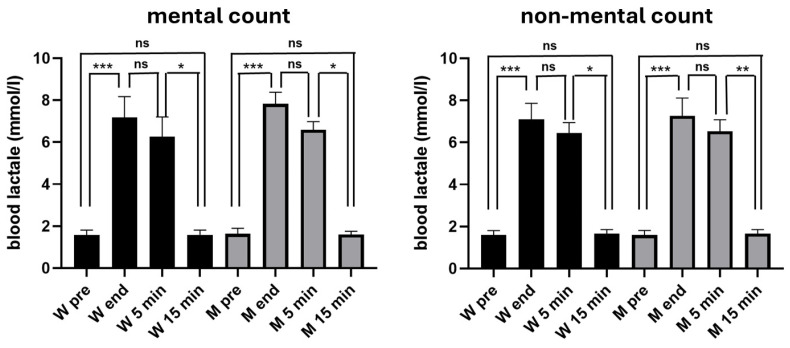
Blood lactate values exhibited by the 12 women (W) and 12 men (M) participating in the present study. The graph on the left shows results obtained when participants had to estimate time by mental counting, and the graph on the right when mental counting was not allowed. In both cases, blood lactate mean values (±SD) measured before the exercise (pre), at its conclusion (end), and 5 min and 15 min after its end are illustrated. Symbols from ANOVA with Dunn’s multiple comparison test: ns, not significant; * *p* < 0.05; ** *p* < 0.01; *** *p* < 0.001.

**Figure 3 jfmk-10-00035-f003:**
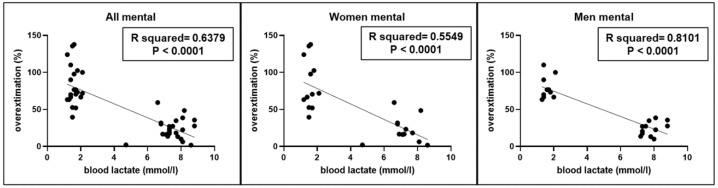
Correlations between blood lactate levels and overestimation (expressed in percent) of 1–5 s time intervals in the whole sample (All) as well as in women and men using mental counting.

**Figure 4 jfmk-10-00035-f004:**
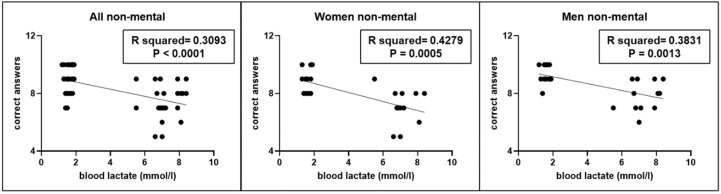
Correlations between blood lactate levels and number of correct answers evaluating difference between two short time intervals (≤5 s), measured in the whole sample (All) as well as in women (W) and men (M) without the possibility of using mental counting.

**Figure 5 jfmk-10-00035-f005:**
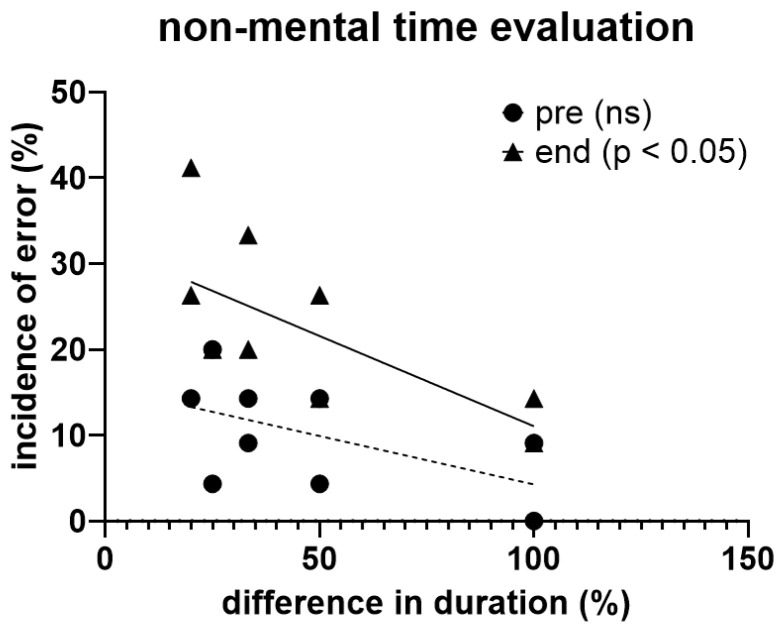
Correlations between the incidence of error (expressed in percent) and difference (expressed in percent) in the duration of the two visual stimuli used for test, before (pre) and at the end of the exhaustive exercise in the whole sample.

## Data Availability

The original contributions presented in this study are included in the article and Appendix A. The data that support the findings of this study can be obtained by the corresponding author upon request.

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
