# Peer review of "Time Estimation Following an Exhaustive Exercise"

_jfmk, 2025, doi:10.3390/jfmk10010035_

Round 1

Reviewer 1 Report (Previous Reviewer 3)

Comments and Suggestions for Authors

The authors have responded one by one to all the recommendations raised

I have no further suggestions on the manuscript 

Thank you.

Author Response

Reviewer 2 Report (Previous Reviewer 2)

Comments and Suggestions for Authors

I do not have any comment.

Author Response

Reviewer 3 Report (Previous Reviewer 1)

Comments and Suggestions for Authors

The conclusion should state the utility/applicability of the obtained findings.

Author Response

This manuscript is a resubmission of an earlier submission. The following is a list of the peer review reports and author responses from that submission.

Round 1

Reviewer 1 Report

Comments and Suggestions for Authors

ABSTRACT

Please, state the mean age of the sample.

A clear conclusion is lacking.

INTRODUCTION

Lines 59-60. Please, provide an explanation by which physical activity affects time perception.

The relationship between time perception and lactacte needs to be more elaborate.

METHODS

How the participants were recruited? Where did they come from?

Line 96, there is a typo, “each subject”. Also a parenthesis is missing.

RESULTS

It is not clear whether there were significant differences by sex regarding lactate levels.

Too many figures. A table including main characteristics of men and women, as well as lactate levels could be of help.

DISCUSSION

The first paragraph is out of place. It should be placed at the end of the section as a limitation.

References 23,24. Please, explain both investigations, specially the exercise proposed and provide an explanation of why a reduction in perceived time was observed.

The authors cite several studies with a smilar topic (11,25,27). It is not clear what are the main differences between said studies and the present one. A deeper description of these investigations is needed.

Lines 290 onwards. In my opinion, these two models should be described in the introduction section.

Limitations as well as the utility and interest of the observed findings should be acknowledged in this section.

CONCLJUSION

A deterioration in time perception due to fatigue is expected as the authors themselves describe throughout the manuscript. Therefore, the conclusion should be more focused on the novelty of the findings and its applicability to the field of research.

Comments on the Quality of English Language

The manuscript should be revised, Some sentences are difficult to read.

Reviewer 2 Report

Comments and Suggestions for Authors

First of all, thank you for submitting to JFMK. The topic of the study is fresh, but the experiment is very simple and the analysis is also simple.

However, the most important thing is “Why was this study conducted?”

I can’t figure out what this study means in terms of sports science.

There is no clear purpose in the abstract or introduction about why this study was conducted.

Please supplement the document so that the meaning, value, and uniqueness of this study can be highlighted.

Title: I hope a few more words can be added so that the overall image of this study can be recalled.

abstract

The background of the study should be briefly mentioned. This will soon become the meaning of this study.

Introduction

The writing of the document is good, but it cannot be said to be an organized and systematic paragraph structure for the purpose of this study. Furthermore, the author conducted a very interesting study, but it is a document that is difficult for readers to understand if they do not have background knowledge. So, I would like to ask the author. I hope this document can be written more logically and easily.

In particular, the last part of the introduction should be clear about the purpose of this study, and it is recommended that a hypothesis be written. Currently, the author's document ends ambiguously at the end of the introduction.

Discussion

The beginning part does not need to repeat the contents written in the research method and introduction. It should proceed in the order of briefly writing about the significance of this study and comparing the results of this study with other previous studies.

How can the results of this study be applied and applied in the field?

There are results comparing men and women, but they are not mentioned in the discussion.

Why is “perception of time intervals between 1 and 5 s” important? There is no basis for speculating about this part.

Looking at the graph of lactate, the points are bifurcated into extremes. Why is this so?

Overall, I hope that the following things will be written in detail in the discussion about similar studies, previous studies, related basic studies, and physiological backgrounds for the experimental results.

Reviewer 3 Report

Comments and Suggestions for Authors

Title

The title is not very explicit and too broad; it should be narrowed to avoid ambiguity.

Introduction

This excessively short section justifies both the purpose of the study and its subsequent applicability in more detail.

The exercise should have a broader justification by adding multiple existing references to vigorous and extensive exercises.

Methods

The fitness level of the subjects is very relevant and should be commented upon, and the response to high intensity is very different.

The final intensity and duration of the exercise depended on the level of the subject. An effort perception scale could have been added because lactate production was blocked earlier in untrained subjects.

Results

VO2max percentages are not indicated, so it is not known if they exceed the maximum lactate steady state (in either condition).

Discussion

As this study indicates, lactate cannot establish the relationship sought in the hypothesis, so it is crucial to add other monitoring.

The discussion is lacking in comparisons with other studies and should delve deeper into the applicability it has

The conclusions should again be warned that the study cannot be generalized because of the small sample size.

A section on limitations should be included